# Nitrogen starvation reveals the mitotic potential of mutants in the S/MAPK pathways

Rostyslav Makarenko [1,2,3], Claire Denis[1,3], Stefania Francesconi [1], Serge Gangloff [1] & Benoît Arcangioli [1✉]

The genetics of quiescence is an emerging field compared to that of growth, yet both states generate spontaneous mutations and genetic diversity fueling evolution. Reconciling mutation rates in dividing conditions and mutation accumulation as a function of time in non-dividing situations remains a challenge. Nitrogen-starved fission yeast cells reversibly arrest proliferation, are metabolically active and highly resistant to a variety of stresses. Here, we show that mutations in stress- and mitogen-activated protein kinase (S/MAPK) signaling pathways are enriched in aging cultures. Targeted resequencing and competition experiments indicate that these mutants arise in the first month of quiescence and expand clonally during the second month at the expense of the parental population. Reconstitution experiments show that S/MAPK modules mediate the sacrifice of many cells for the benefit of some mutants. These findings suggest that non-dividing conditions promote genetic diversity to generate a social cellular environment prone to kin selection.

[1] Genome Dynamics Unit, Institut Pasteur, UMR3525 CNRS, 75015 Paris, France. [2] Sorbonne Université, École Doctorale 515, 75005 Paris, France. [3]These authors contributed equally: Rostyslav Makarenko, Claire Denis, Stefania Francesconi. ✉email: benoit.arcangioli@pasteur.fr

In nature, cells alternate between replicative and non-replicative states depending on the fluctuating environmental conditions and physiological requirements[1]. The reversible non-replicative state resulting from cell division arrest or transition to a more specialized cellular type is predominant in nature and termed quiescence or G0. The transition between replicative and non-replicative states is interpreted by the cell as a stress that requires a quick response for survival. Therefore, the cells have evolved mechanisms to cope with the perpetual fluctuation of their habitat through optimized genetic and epigenetic regulations that allow a rapid metabolic adaptation, coordinate developmental programs, and tissue homeostasis to ensure longevity.

Several aspects of quiescence have been studied in both budding (Saccharomyces cerevisiae) and fission (Schizosaccharomyces pombe) yeast. These include a decrease in transcription, translation and metabolism, cell cycle arrest, activation of autophagy, an increase in the resistance to a large variety of stresses[2], as well as a variation in telomere length and in RNA interference functions in S. pombe[3–5].

In the laboratory, chronological lifespan is traditionally studied by measuring the viability over time of cells having reached the stationary phase after exhaustion of the carbon source from an otherwise nitrogen-rich medium[6,7]. The stationary phase by glucose exhaustion[8,9], as well as starvation by iron[10], sulfur[11], phosphate[12], or transfer to water[13] is rarely used to study long-term quiescence in fission yeast because cells die rapidly. Alternatively, nitrogen starvation has been extensively used to study mating, meiosis, and quiescence in fission yeast[13,14]. Under these conditions, the TOR (target of rapamycin) and the SAPK (stress-activated protein kinases) pathways set off two rapid cell divisions with no cell growth to arrest cells in G1 and promote the expression of pheromones[2,13–16]. When both mating-type partners are present, the mating and the meiotic programs are engaged, whereas in the absence of a partner cells enter G0[2,15]. The quiescent state can be sustained for prolonged periods and provides a unique opportunity to analyze the outcome of time-dependent processes involved in quiescence entry, maintenance, and exit[14]. Thus, nitrogen starvation is a simple system to induce differentiation and a valuable complementary approach to study biological aging in quiescence[17].

The S/MAPKs (stress or mitogen-activated protein kinases) signaling pathways are required for sensing and responding to internal and external stimuli and various environmental stress. This fast response involves a cascade of kinases that controls basic biological processes[18]. The SAPK module is composed of the MAPKKKs Win1 and Wis4, the MAPKK Wis1 and MAPK Sty1/Spc1. Early work has identified Sty1 and Wis1 as essential genes to arrest the cell cycle in G1, enter in quiescence, and maintain viability[16]. The MAPK module, required for cytokinesis and cell wall integrity, is composed of Mkh1 (MAPKKK), Pek1 (MAPKK), and Pmk1 (MAPK)[19]. Although it was reported that Pmk1 participates in Sty1 activation of the downstream effector Atf1 (ref. [20]), the crosstalk between the SAPK and MAPK modules is not fully understood. Unfortunately, the inability of the SAPK deletion mutants[2] to efficiently enter G0 has precluded from addressing whether the SAPK pathways participate in stress resistance during quiescence as well.

During growth, spontaneous mutations accumulate as a function of cell division[21] or DNA replication[22]. Under growth conditions, genome-wide sequencing approaches have shown that the mutation rate is in the range of $2 \times 10^{-10}$ mutation per nucleotide and per generation in fission yeast[23,24]. Extensive work in many species has shown a similar mutation spectrum where single-nucleotide variants (SNVs) are more frequent than insertions and deletions (indels) during growth[25]. In quiescence, DNA damage is efficiently repaired indicating that some repair pathways are functional[11,26] and we recently showed that mutations increase linearly as a function of time, starting on day 1 (ref. [27]). When the medium was replaced every other week to avoid the accumulation of nitrogen released by dead cells, we found a random distribution of mutations that reached 0.6 mutations per genome after 3 months of quiescence, with about one half of the strains exhibiting no detectable mutation. In addition, and unlike cycling cells, SNVs and indels accumulate at the same pace and deletions dominate insertions, a signature that we called chronos[27]. Thus, the genetics of quiescence fulfills the experimental conditions to select for de novo mutations that sustain viability in non-dividing conditions and explore a different mutation spectrum than screens using deletion libraries or conditional mutants[28].

The goal of this study is to select for spontaneous mutants appearing in nitrogen-starved fission yeast cells that are capable of resuming growth after 2 and 3 months of quiescence. Using WGS, we find recurrent de novo mutations in genes directly related to the S/MAPK pathways. Targeted resequencing and competition experiments indicate that these mutants expand clonally at the expense of the parental cell population. Thus, we uncover an unexpected scavenging function that mediates the sacrifice of many cells for the benefit of some mutants during quiescence. Comparisons between the genetics of quiescence of bacteria, yeast, and mammalian cells are discussed.

## Results

**The S/MAPK mutants are enriched among survivors**. To explore the potential of the genetics of quiescence we grew a haploid prototrophic strain (Table 1) in minimum medium (EMM) for 27 generations, enough to produce $10^8$ cells and limit the occurrence of mutations. The culture is then transferred to minimum medium without nitrogen at low cell density ($10^6$ cells per mL) and kept in non-proliferative conditions for 3 months without changing the medium (Methods). Based on previous work establishing somatic and quiescence mutation rates[23,24,27] and our experimental settings reducing somatic mutations, we anticipated 0.6 mutations per genome, generated mainly during the 3 months of starvation. The viability curve (Fig. 1a) shows three phases; during the first 20 days, the viability is maintained (phase I), followed by a rapid decline (phase II), before slowly stabilizing (phase III). Fluorescent-activated cell sorting (FACS) analysis of PB1623 indicates that cells with no DNA accumulate over time (Fig. 1b). The fraction of dead cells correlates with the

| Table 1 Strains used in this study. | |
| --- | --- |
| **Strain** | **Relevant Genotype** |
| PB1623 | Msmt0 |
| 972 h⁻ | h⁻ |
| PB2262 | Msmt0 SPBC31A8.02::mCherry::KanR |
| PB2740 | Msmt0 win1-394 +13 bp SPBC31A8.02:: mCherry:KanR |
| PB2741 | Msmt0 mkh1-2929 −10 bp SPBC31A8.02::mCherry::KanR |
| PB2850 | PΔ17 sty1-G197T SPBC31A8.02::mCherry::KanR |
| PB2851 | Msmt0 sty1-C74G SPBC31A8.02::mCherry::KanR |
| PB2739 | Msmt0 sgf73-896+T SPBC31A8.02::mCherry::KanR |
| PB2532 | Msmt0 sgf73-896 +T |
| PB2921 | Msmt0 sty1-G197T |
| PB2926 | Msmt0 sty1-C74G |
| PB2928 | Msmt0 pmk1-153 +A |
| PB2310 | Msmt0 sty1Δ::ura4+ |
| PB3242 | Msmt0 mkh1-640 +A |
| PB2932 | Msmt0 win1-1086 −G |
| PB2933 | Msmt0 win1-394 +13 bp |
| PB2457 | PΔ17 win1Δ::KanR |

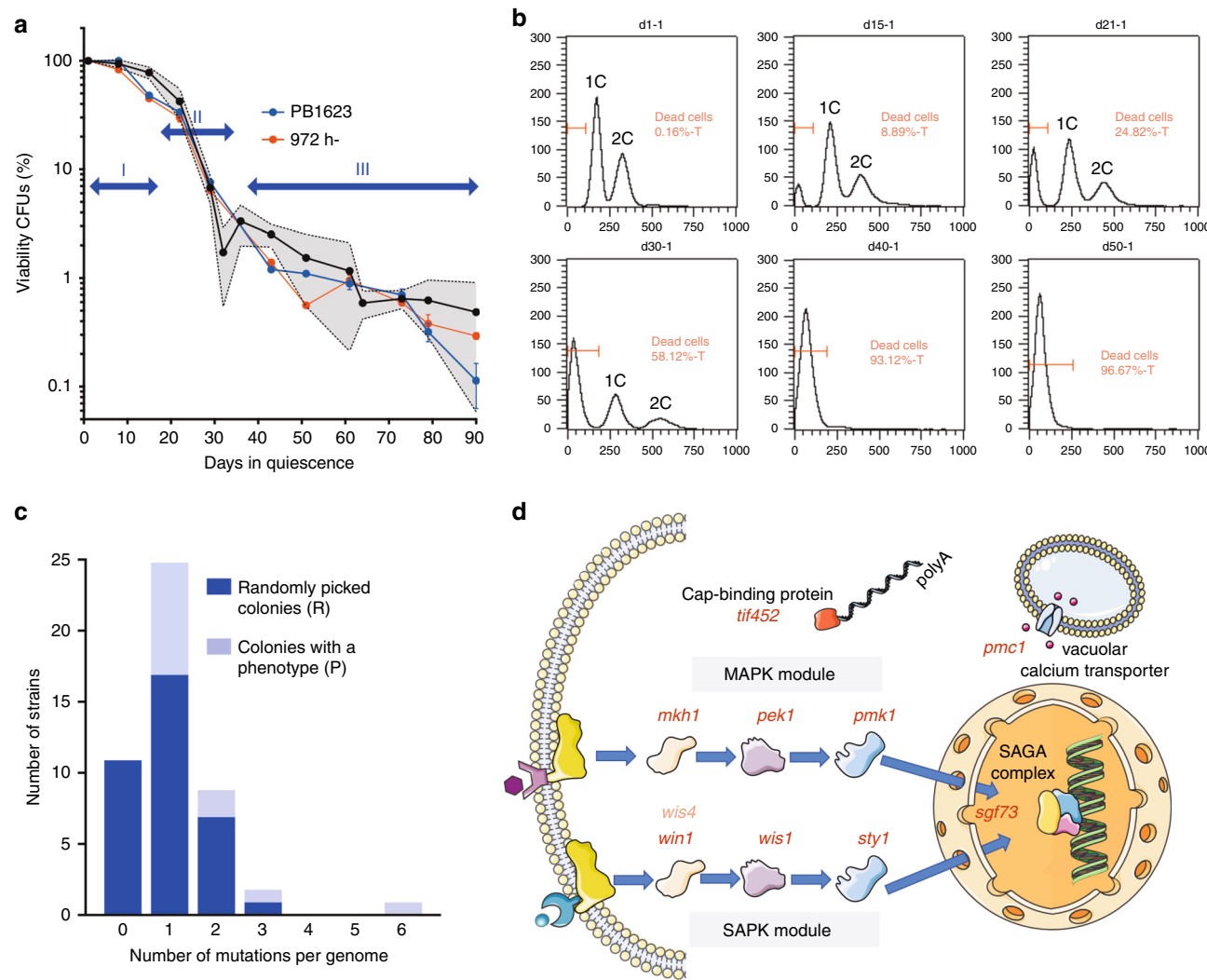

**Fig. 1 Viability and distribution of mutations in nitrogen-starved cultures. a** The ability to form colonies (CFUs) was monitored on 17 independent cultures for 3 months by plating aliquots of the cultures on rich medium at various time points (black curve). The mean is represented and the gray area delineates the limits of the standard deviations. The red (wild-type reference strain 972 h⁻) and blue (our wild-type progenitor PB1623) curves were made in parallel on a culture that was split into three on day 1 of quiescence. The CFUs were determined in triplicate at days 21, 61, 73, 79, and 90. The three phases are highlighted. Source data are provided as a Source Data file. **b** Analysis of the DNA content of PB1623 cells determined by FACS at 1, 15, 21, 30, 40, and 50 days of starvation. **c** Distribution of the number of mutations per genome after 3 months of starvation in the 36 randomly picked (R) colonies and in the 12 colonies selected for their sensitivity to various agents. **d** Schematic representation of the pathways in which the nine genes analyzed in this study operate.

capacity of the culture to form colonies (CFUs). When the medium was replaced every other week[27], phase III was not observed and cells continued losing viability. The presence of phase III is also observed in the unrefreshed culture of the 972 h⁻ reference strain (Fig. 1a, Table 1). This indicates that phase III depends on the experimental conditions and not on the genetic polymorphism (Supplementary Table 1) between the two strains. Thus, the presence of phase III suggests a phenotypic adaptation of the cells in the surviving population. Accordingly, when we monitored the sensitivity of the survivors to conditions affecting a broad range of cellular functions, we found 30 clones out of 150 (20%) exhibiting a phenotype after 3 months (Methods and Supplementary Table 2). Genetic backcrosses followed by tetrad analysis indicate that the 30 strains exhibit a Mendelian 2:2 segregation of the phenotype indicating a defect at a single locus for each strain (Supplementary Fig. 1a, b). This elevated level of surviving cells exhibiting a sensitivity phenotype was not expected with the anticipated level of mutation and suggests a strong gain of fitness.

To estimate the accumulation of mutations and identify some of the genes mutated, we sequenced the genome of 12 strains selected for their sensitivity to various conditions as well as of 36 survivors picked randomly. We found 22 mutations in the genome of the 12 clones selected for their phenotype and 8 out of 12 strains contain a single mutation indicating that this mutation is likely responsible for the phenotype (Fig. 1c, Supplementary Methods). Although one strain contains four mutations in a mischaracterized gene (SPCC569.02c-antisense-1), mutations in clusters are rare but have been previously observed[29]. We also identified 36 mutations in the 36 strains picked randomly, among which 16 are in genes belonging to the S/MAPK pathways[16,19], to their downstream targets eIF4E/CAP-binding factor Tif452 (ref. [30]) and Sgf73 coding for a subunit of the SAGA transcriptional complex[31,32]. We also found Pmc1, the vacuolar calcium transporter, that is remotely related to the S/MAPK pathways and whose absence makes cells sensitive to calcium and rapamycin under salt stress[33,34] (Fig. 1c, d, Supplementary Tables 3 and 4).

Other mutations in genes including *cut1*, *gap1*, *papb*, and *pfl4* were also identified, but were not further studied in this work (Supplementary Table 3). The extent of mutations present after 3 months of quiescence is about two-fold that previously reported, and their spectrum is consistent with chronos[23,27].

**Targeted resequencing of long-term survivors**. The number and frequency of alleles that we found prompted us to analyze the occurrence of S/MAPK mutations over time using a deep targeted sequencing approach of eight genes identified above; *wis1* was not included in this round (Fig. 1d and Methods). We first analyzed the culture from which the mutants were isolated (thereafter called culture 0) along with three additional independent cultures (1–3) on day 1 and after 2 and 3 months of quiescence, providing a readout of the dynamics of the S/MAPK mutants in the cell population (Fig. 2a left, Methods).

Mutations in the targeted genes (Fig. 1d) are detected in the four cultures and account for 17.7–64.7% of the surviving cell population (Fig. 2b, c, Supplementary Table 5, Methods and Supplementary Methods), whereas none are detected above 0.1% with a coverage of at least 5000 on day 1. Between 2 and 3 months, the distribution of alleles in each culture is fluctuating. For example, in culture 2 the *sgf73*-896 +T allele represents 9.6% of the population at 2 months of starvation and reaches 43.5% a month later, indicating cell division, at the expense of *mkh1*-640 +A. Conversely, in culture 3 the proportion of *win1* alleles remains stable over the same time frame. These observations indicate that each mutant behaves individually in the population, and that clonal interference can affect the ability to undergo cell division and impact the fitness of the alleles (Fig. 2b, c). The *sgf73*-896 +T and *mkh1*-640 +A alleles are found in several cultures and the mutations are located in stretches of 8 As and 7 As, respectively. The mutation *win1*-1273 +17 bp, previously reported[35], and *win1*-394 +13 bp are direct-repeat duplications of 17 and 13 bp, respectively (Fig. 3b, Supplementary Table 5). To validate our calling procedure, we went back to the frozen samples (Methods) to isolate several clones exhibiting a phenotype and confirm by Sanger sequencing the presence of the corresponding mutations (alleles in red in Fig. 2, Supplementary Table 5 and Methods).

**Mutations occur in nitrogen starvation**. The large proportion of indels is reminiscent of the chronos signature[27] and supports the view that many mutations arise during prolonged nitrogen starvation. We reasoned that the mutations arising during growth will be maintained in similar proportions over time. Thus, if a culture is divided on day 1 of quiescence into subcultures, it should be possible to distinguish preexisting mutations that will be present in several subcultures from unique mutations that have occurred independently during the quiescence period. To address this issue, we grew a culture from a single fresh colony that was transferred to EMM-N for 1 day prior to being dispatched into six subcultures containing $10^7$ cells each (Fig. 2a, right panel). In this experiment, we analyzed one thousand colonies from day 1 before splitting and from each subculture at 2 months of starvation. The nine genes, including *wis1*, were PCR amplified in duplicate from each subculture and were Illumina sequenced separately. Thus, the sensitivity of our calling procedure was improved by increasing the coverage and collecting the cells only at 2 months (Methods). Like in the four independent cultures, the proportion of mutations in the targeted genes in the six subcultures ranged from 19.5% to 50.9% (Fig. 2d, Supplementary Table 5) at 2 months, whereas no mutation was detected above 0.1% on day 1. The most frequent alleles are *sty1*-C545T and *sty1*-A553T and are restricted to subcultures 5 and 2, respectively. Several hotspot

mutations found in the previous experiments (*mkh1*-640 +A, *win1*-1273 +17 bp, and *win1*-394 +13 bp) are found again in some but not all subcultures. If we exclude the hotspots, the vast majority of the other alleles in each subculture are unique, strongly supporting the model in which the largest part of the mutations arises independently during nitrogen starvation (see Discussion).

**Stress resistance in quiescence requires the SAPK pathway**. Using the approach described above, we identified 195 independent mutations in nine genes (Fig. 3a). When their spectrum was analyzed, different distributions of variants were observed (Fig. 3a, b). While frameshift indels or nonsense mutations are found in the MAPK genes, it is remarkable that the genes in the SAPK alleles only contain missense or in frame indels, with the notable exception of the Win1 upstream kinase. However, this latter function could be compensated for by Wis4, a related MAPKKK (Fig. 1d). These data strongly suggest that some functions of the SAPK pathway cannot be eliminated in quiescence.

Since it is well established that *sty1*Δ deletion strains enter and survive poorly in quiescence[16], we tested whether the 2 *sty1* alleles identified by sequencing the 48 strains (Supplementary Table 3) are proficient to enter G0.

We found that both alleles are capable of entering quiescence although slower than wild-type (Fig. 4a, b). This unique property prompted us to address whether the SAPK pathway is also required to respond to stress during quiescence, and found that the *sty1* mutations are sensitive to temperature and $H_2O_2$ both in vegetative and quiescent cells (Fig. 4c–f). The comparison with the deleted strain, which is only possible in growing cells, supports the prediction that these alleles are hypomorphic (Fig. 4c–f). On the contrary, as predicted by their mutation spectrum, the *win1* alleles behave like the deletion (Fig. 4c–f). These results indicate that the SAPK pathway is required to confer the widely conserved elevated stress resistance observed at least in early quiescent cells[14], but hampering its function provides an advantage upon exit from prolonged starvation.

**Growth advantage under nitrogen starvation**. To determine the advantage conferred by the mutants in nitrogen-starved conditions, we first compared the viability of the wild-type parental strain with several mutants in the S/MAPK modules during long-term starvation. The survival profile of *mkh1*-2929 −10 bp and *sgf73*-896 +T mutant strains is similar to the parental wild-type strain while that of *win1*-394 +13 bp, *sty1*-C74G and *sty1*-G197T decreases rapidly until it roughly reaches 1% (Fig. 5a, Table 1). These data suggest that the advantage may only manifest itself through cooperation in a heterogeneous cellular population[36]. To test this idea, we set up competition experiments using cultures containing the wild-type and one of the mutant strains, as previously performed in bacteria to define the Growth Advantage in Stationary Phase (GASP) phenotype[37]. After 1 day of quiescence, we mixed 99.9% wild-type and 0.1% kanamycin marked strains (Methods, Table 1). In the co-culture conditions, the *win1*-394 +13 bp, *sty1*-C74G, *sty1*-G197T, *mkh1*-2929 −10 bp, and *sgf73*-896 +T marked mutant strains reveal their mitotic potential. During the initial 20–30 days, the mutants die with kinetics similar to that of the individual cultures (Fig. 5b). Subsequently, the mutants start to proliferate exponentially with a generation time of about one division every 3 days. The SAPK mutants end up forming a population larger than their initial inoculum to outcompete the wild-type at 2 months of starvation (Fig. 5b). The simplest hypothesis to account for this result is that the progressive death of the cells (mainly the wild-type

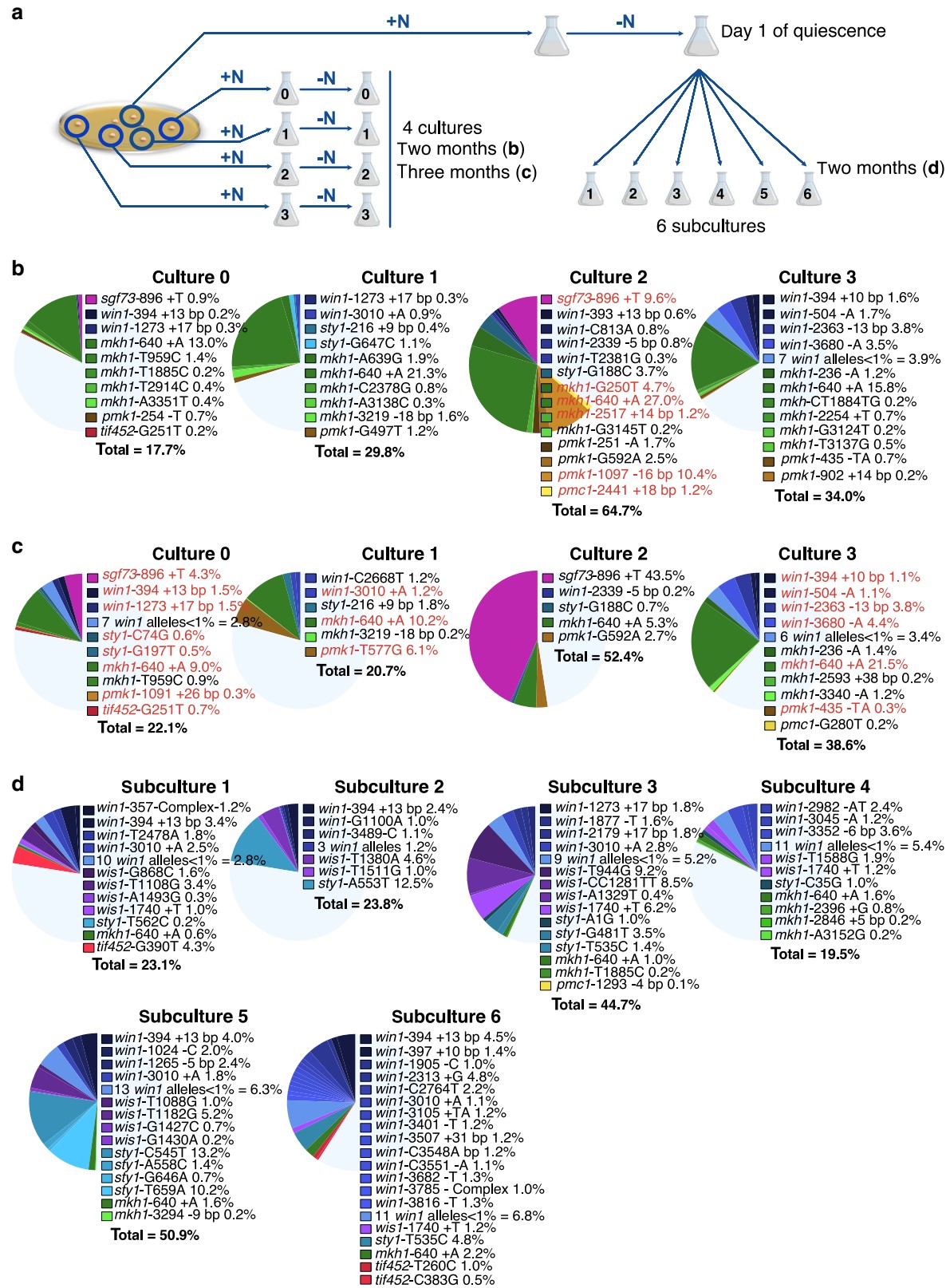

**Fig. 2 Targeted resequencing of genes in the S/MAPK pathways. a** Schematic view of the strategy used to obtain nitrogen-starved cultures. The same progenitor strain was used to grow (+N) three independent cultures (in addition to culture 0) prior to shifting to quiescence (−N) (left panel). The same approach was used to grow a single culture prior to splitting on day 1 of quiescence into six parallel subcultures (right panel). Samples were analyzed after 2 and 3 months of starvation. **b** Distribution and percentage of mutations found in the thousand colonies picked from cultures 0 to 3 after 2 months. The alleles in red were confirmed by Sanger sequencing the stress-sensitive colonies found among 120 cells plated out from the frozen pools. **c** same as **b** after 3 months. **d** Distribution and percentage of mutations in the subcultures after 2 months of starvation.

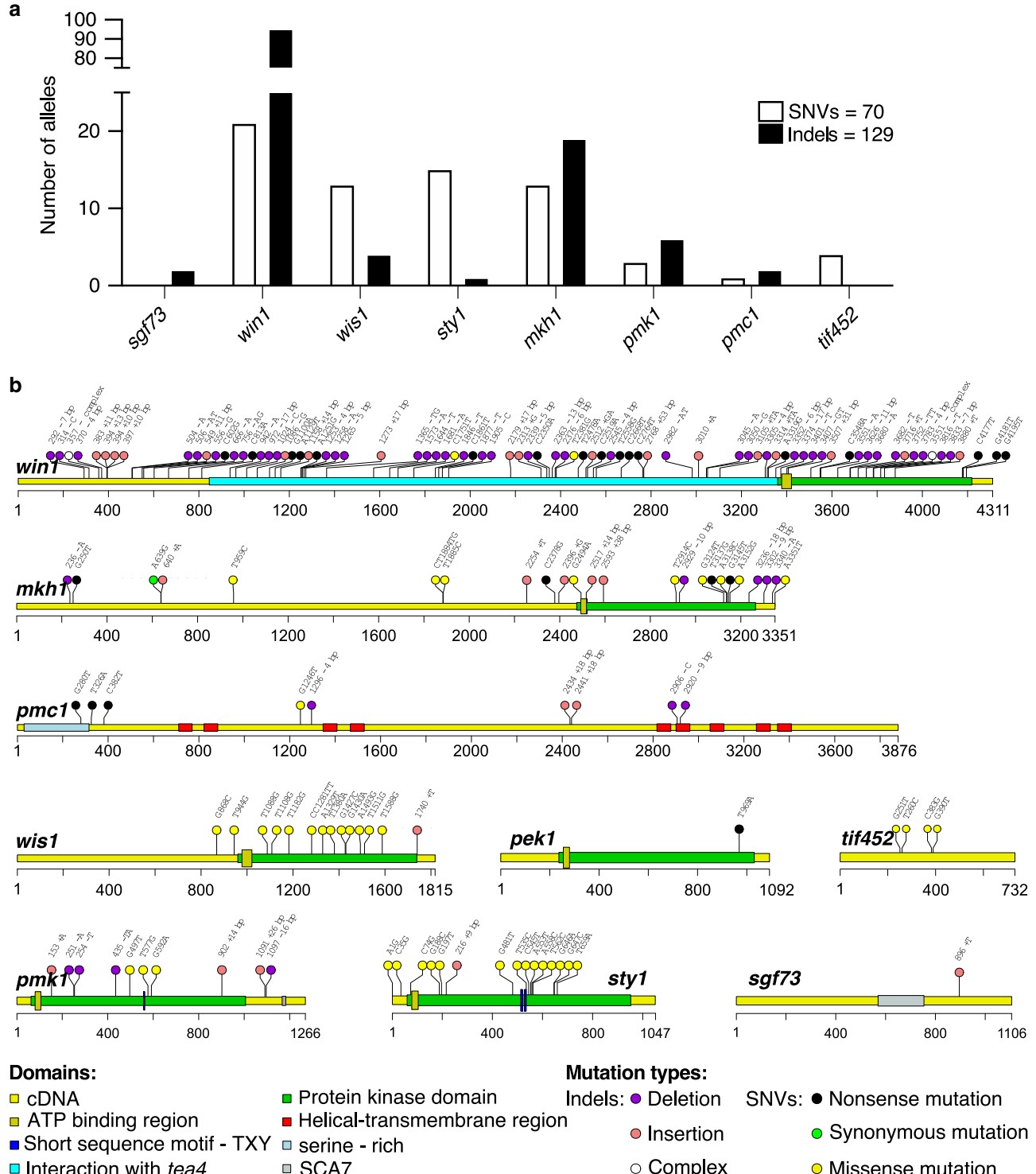

**Fig. 3 Type and position of the mutations found in the genes of interest. a** Distribution of the number of SNVs and indels found among the 199 alleles identified by targeted resequencing of cultures 1–3 and subcultures 1–6. In cultures 1–3, mutations found at both 2 and 3 months were counted only once. **b** Mutations in the nine genes of interest identified by WGS and by targeted resequencing. Known domains and mutation types are indicated.

progenitor) generates a release of nitrogen that primarily benefits the mutant cells. However, the nitrogen concentration released by the dead cells in long-term starved cultures is fluctuating because it is constantly metabolized by viable cells (Supplementary Fig. 2). This hypothesis would also explain why the single cultures of most mutants do not show growth. In this case, nitrogen traces released by dead cells are shared by too many identical mutants, a

situation radically different from mutants arising stochastically and progressively.

**Scavenging behavior and kin selection**. In fission yeast, minimum medium is supplemented with 5 g/L (34 mM) of glutamate or ammonium chloride (93.5 mM) while rich medium contains 5 g/L of yeast extract. These concentrations allow reaching $10^8$ and

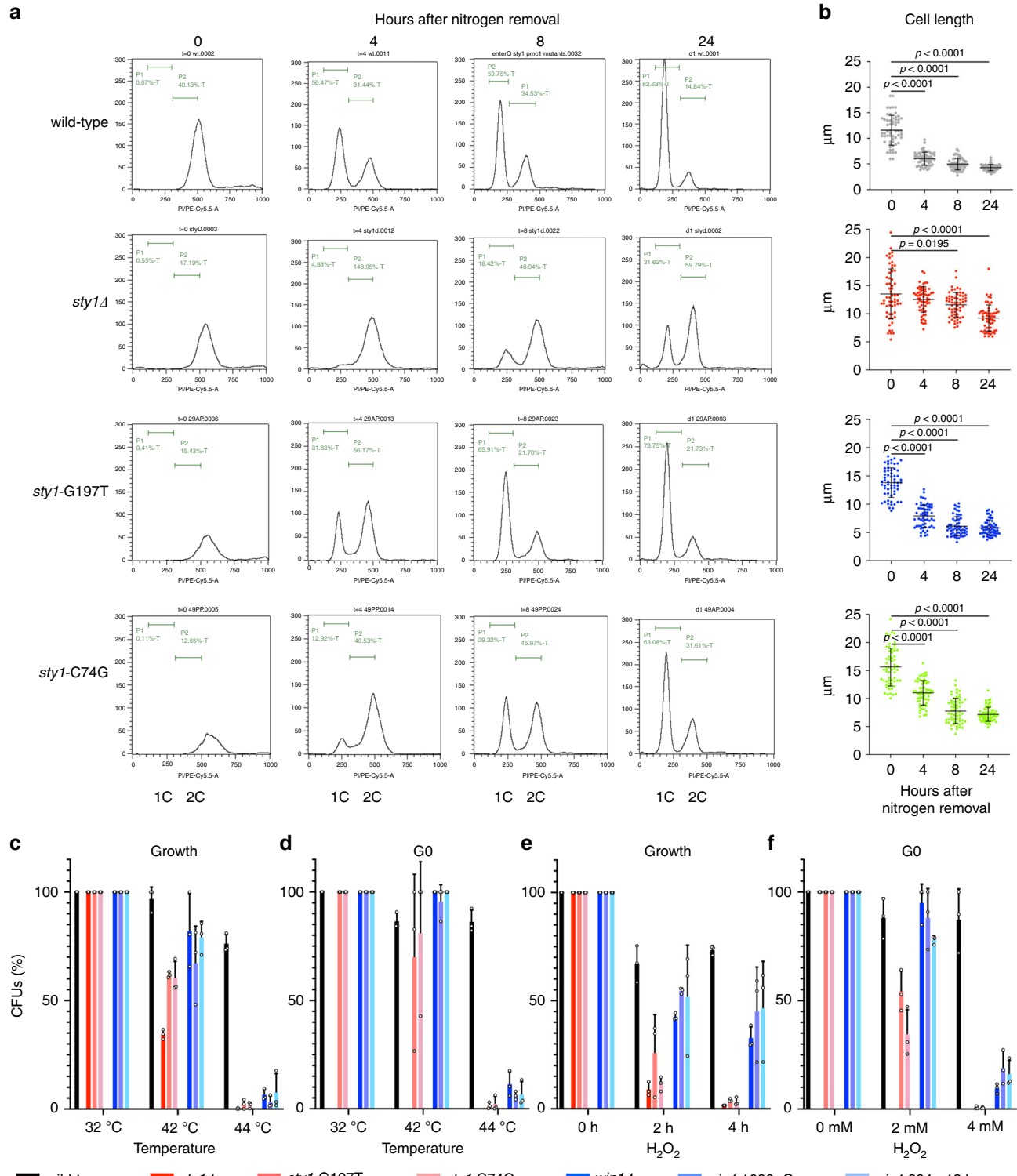

**Fig. 4 Analysis of quiescence entry and sensitivity to stress. a** DNA content analysis after nitrogen removal was determined by FACS. **b** Cell size determined with ImageJ on pictures ($n = 60$) after nitrogen removal. The mean and SD are depicted. Two-tailed Mann–Whitney tests were applied and only significant exact $p$ values are shown. Source data are provided as a Source Data file. **c** The growing strains were incubated for 2h at the indicated temperature prior to plating. **d** Viability of 1-day-old quiescent strains incubated for 2 h at the indicated temperature prior to plating. **e** Viability of the growing strains following exposure to 1mM $H_2O_2$ for the indicated times. **f** Viability of 1-day-old quiescent strains following exposure for 2 h to the indicated amounts of $H_2O_2$ prior to plating. After plating on rich medium and incubating at 32 °C for 3 days, colony-forming units were scored. The values determined at 32 °C, or in the absence of treatment, were set to 100%. The mean, SD, and individual values ($n = 3$) are shown. Source data are provided as a Source Data file.

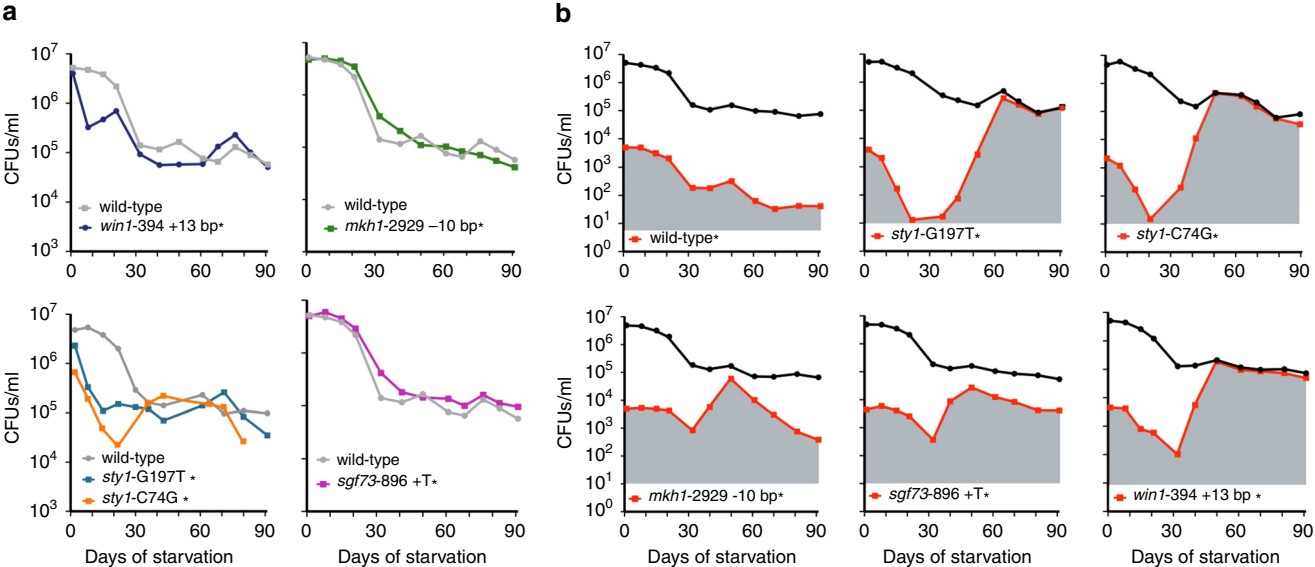

**Fig. 5 Survival behavior of the mutants depends on the environment. a** Survival of single mutants (colony-forming units) with their wild-type control during long-term starvation. The star (*) indicates the presence of the kanamycin marker in the background. Source data are provided as a Source Data file. **b** Co-cultures where the wild-type is contaminated at day 1 of quiescence with 0.1% of the indicated strain (*) marked with the kanamycin gene (red). The black curve indicates the viability of the entire population, while the red one reveals the colony-forming units of the kanamycin-resistant strain in this population. Each point corresponds to the mean of independent cultures ($n = 2$), except for the *sty1* mutants that were only tested once. Source data are provided as a Source Data file.

$10^9$ cells per mL, respectively. To investigate the possibility of a scavenging behavior, we first established by FACS analysis the minimal concentration of glutamate necessary to engage the first S phase is 0.03 mM and that to complete one replication round with no increase in cell number is between 0.09 and 0.12 mM for $4 \times 10^6$ wild-type cells per mL (Supplementary Fig. 2a). Thus, to mimic a slow nitrogen release during an extensive quiescence period, we provided traces of nitrogen by adding either glutamate (0.02 mM), ammonium chloride, or yeast extract to 3-day-old quiescent cultures (day 0) twice a day for up to 3 days after the first addition (Fig. 6, Methods). The mass ($OD_{600}$), ability to form colonies (CFUs), DNA content by FACS and presence of septa by Calcofluor white staining of wild-type and three mutated downstream players of the S/MAPK pathways (*sgf73*-896 +T, *sty1*-G197T and *pmk1*-869 +A strains) were followed daily (Fig. 6, Supplementary Fig. 3). All the strains increased their cell mass with time, but with no increase in cell number (Fig. 6a, Supplementary Table 6). While the mutants retain their initial number of viable cells (CFUs), the wild-type counterpart progressively forms fewer colonies (Fig. 6a). The analysis of the DNA content indicates that the wild-type cells engage DNA replication with the subsequent appearance of a propidium-negative peak indicative of nuclear DNA degradation (Supplementary Fig. 3, Fig. 6b). Calcofluor white staining reveals the septum of dividing cells. In addition, a large proportion of cells in the wild-type population becomes brightly stained on day 3 after the first addition regardless of the nitrogen source, a phenotype not observed in the mutant cells (Fig. 6c). These Calcofluor bright cells are smooth under phase contrast microscopy and are stained neither by propidium iodide nor by Coomassie blue (Supplementary Fig. 4), indicating that they no longer contain any detectable nucleic acids or proteins. Additionally, the increasing amount of Calcofluor bright cells correlates with the proportion of propidium-negative cells observed by FACS (Supplementary Fig. 4) and identify cells incapable of forming colonies.

In this experiment, the two kinase cascades do not exhibit the same response to low doses of nitrogen. While a subpopulation of

*sgf73* and *pmk1* mutants engages DNA replication, mitosis, septation, and prevents cell death (Fig. 6b, c, Supplementary Fig. 3), the *sty1* mutant delays its first S phase and cell division for 3 days after the first addition (Fig. 6b, c), likely waiting for a higher concentration of nitrogen to become available before engaging DNA replication and septation (Supplementary Fig. 2). Thus, a delayed commitment to DNA replication in situations where nitrogen is scarce can confer a survival and proliferation advantage at quiescence exit. The phenotype of both wild-type and mutant strains are independent of the genetic background, since the segregants from two tetrads of the *pmk1* mutant crossed into the 972 h⁻ background exhibit a behavior indistinguishable from that of the strains shown in Fig. 6 upon addition of traces of glutamate (Supplementary Fig. 5). Taken together, these results indicate that a functional SAPK pathway is required to engage rapidly DNA replication upon quiescence exit under suboptimal concentrations of nitrogen, while a proficient MAPK pathway is required for killing the cells that have engaged replication under these conditions. On the contrary, when the concentration of glutamate is brought up to 0.2 mM in a single step in a 3-day-old quiescent culture, all the strains exit quiescence and fully replicate their DNA (Supplementary Fig. 2b), supporting the idea that a slow and progressive release of at least nitrogen by the dying cells promotes the further killing of the wild-type. This scavenging behavior also provides an explanation for the positive clonal advantage of the mutants in the S/MAPK pathways in the presence of parental cells during extended periods of nitrogen starvation.

## Discussion

Using laboratory evolution experiments, we found that mutants related to the S/MAPK response pathway are preferentially recovered upon exit from 2 months of nitrogen-limited conditions and that their appearance becomes predictable. This work is complementary to the previous identification of mutants in systematic screens of deletion libraries coupled with barcode

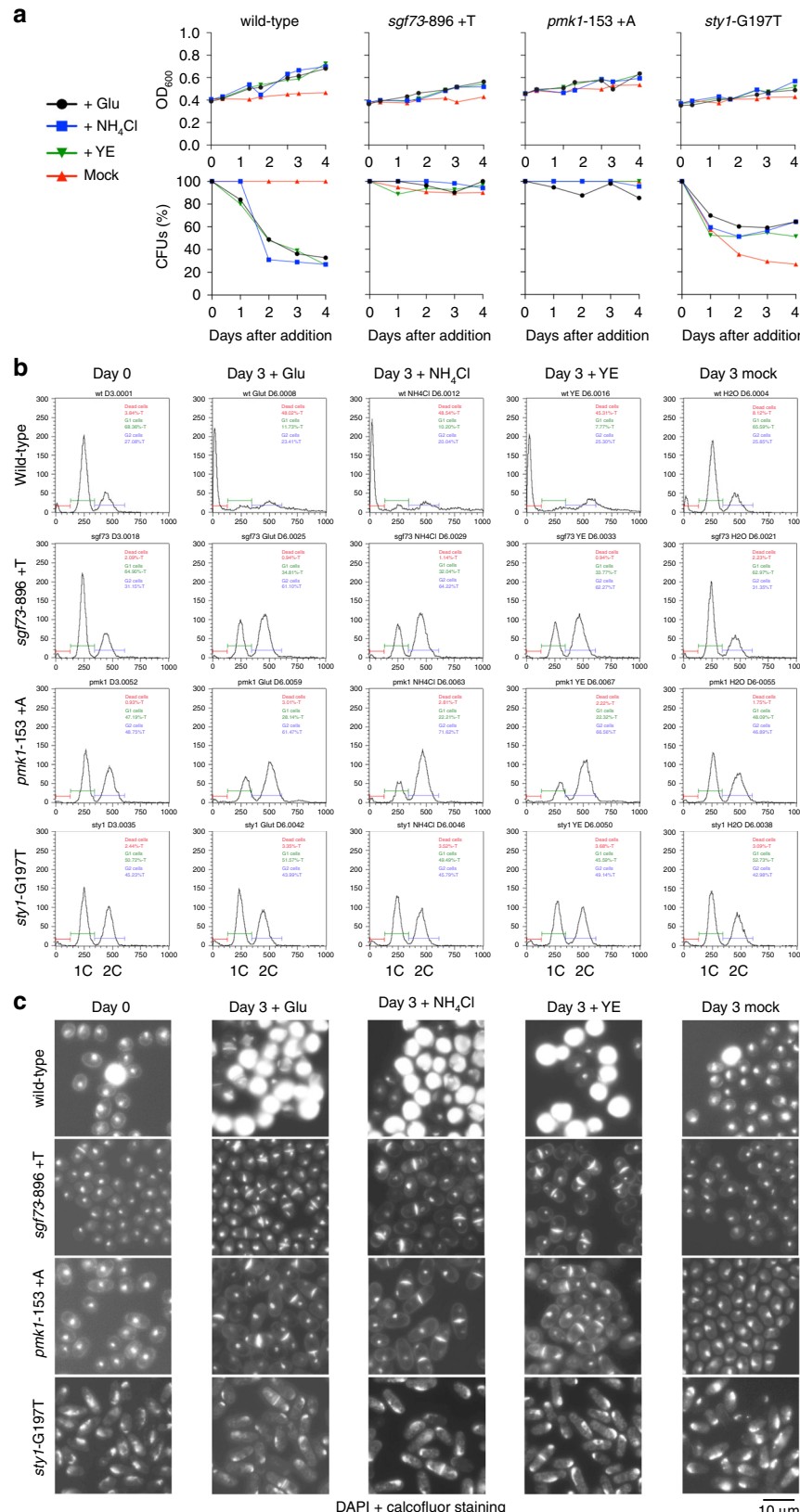

sequencing or of conditional mutants that either sustain quiescence, promote exit, or are resistant to TORC1 inhibition that mimics nitrogen starvation[3,28,38–40]. In the former fission yeast studies, most mutants that have been identified encode proteins associated with membranes and autophagy functions, among which the *pek1* mutant (MAPK module) is the only gene shared with this work[27,28]. In budding yeast, high-throughput screening identified *HOG1*, *SSK2* (SAPK pathway), and *SGF73* in glucose- and sulfate-limited conditions, respectively[32,41]. Recently, *HOG1* was also found mutated in yeast after a 2-year starvation in sealed

**Fig. 6 Scavenging behavior promotes quiescence exit of the S/MAPK mutants.** Glutamate, ammonium chloride, or yeast extract was added twice a day to 3-day-old quiescent cultures (day 0) containing $4 \times 10^6$ cells per mL at a 1/1500 dilution. The same volume of water was added as a control (mock). **a** Cell mass increase ($OD_{600nm}$) (upper panels) was followed up to day 3 after the first addition; CFUs (lower panels) was determined for up to 4 days after the first addition. The graph corresponds to a representative experiment out of three reproducible and independent ones. **b** DNA content analysis was determined by FACS on day 0, prior to nitrogen addition to 3-day-old quiescent cultures, and on day 3 after the first addition. The DNA content by FACS analysis at the intermediate time points of days 1 and 2 after addition are shown in Supplementary Fig. 3. **c** Fixed cells were stained with a DAPI-Calcofluor solution (1 μg/mL DAPI, 1 μg/mL p-phenylenediamine, 25 μg/mL Calcofluor white) to visualize the nuclei and the septum formation on days 0 and 3 after the first addition. The presented micrographs are representative of the situation observed in the field of 20 pictures each from three independent experiments.

bottles of beer[42]. Thus, mutating the stress-response pathway emerges as a common strategy to promote a growth advantage under different starvation conditions in two unrelated yeast, indicating a high degree of converging evolution[43]. Furthermore, our results indicate that mutations in SAPK and MAPK pathways provide a growth advantage through two different modes: mutating the SAPK module delays S phase entry whereas mutations in the MAPK module (*pmk1* and *sgf73*) protect against cell death (Fig. 6, Supplementary Figs. 3–5).

It is counterintuitive that mutations in the stress-mitogen response pathway provide an advantage in starving and slow-dividing cells but other examples have been reported. In several filamentous fungi spore germination is controlled by light, and phytochrome mutants working upstream of the MAP kinase SakA germinate faster than wild-type[44]. In Arabidopsis, the MAPK *mpk6* mutant promotes exit from quiescence of the shoot apical meristem[45]. In human and mouse muscles, p38/Sty1 inhibition promotes the proliferation of quiescent satellite cells[46]. In activated lymphocytes, p38/Sty1 deficiency causes hyperproliferation, correlating with a decrease in eIF4E activity[47]. Thus, the S/MAPK mutants can successfully exit from the quiescent state at a lower threshold of environmental or physiological cues. However, the advantage of a rapid exit from nitrogen-limited conditions probably does not compensate for the pleiotropic negative impact of a defect in the S/MAPK pathways, and such runaway mutations are unlikely to become fixed in the population. For instance, two hotspot mutations in *sgf73* and *mkh1* are located in homonucleotide runs and near low complexity sequences known to be prone to reversion during DNA replication. Finally, the mutations in the nine genes isolated here are not sterile and can therefore be efficiently removed by sexual reproduction with beneficial effects for the population[48].

The molecular mechanism generating mutations during quiescence in fission yeast is not yet understood, but it was proposed to be due to errors during the repair of spontaneous or induced DNA lesions[11,26,27]. Whereas it is difficult to distinguish whether DNA lesions are processed into mutations during quiescence or become fixed upon resuming DNA replication, multiple lines of evidence support the view that the reported mutations arise and are phenotypically revealed during quiescence. First, according to the well-established replication-dependent mutation rate in wild-type cells[23,24], the low number of cell divisions (<27) prior to quiescence entry is insufficient to generate the diversity and number of mutants observed in the cultures, while prolonged starvation generates mutations that reach the observed one mutation per genome at 3 months. Second, although biased by selection, the spectrum of mutations determined by whole-genome sequencing (WGS) and to some extent by targeted resequencing is reminiscent of the chronos signature[27]. Third, the variability in distribution, abundance, and type of mutations observed in the six subcultures derived from the same clonal culture supports independent events arising during starvation. Fourth, the death of the mutants during the first month strongly supports the model in which the mutations

in the SAPK pathway isolated after 2 months of incubation have arisen during starvation and indicates that the time frame of appearance of a mutation as well as its type (SNVs/indels) and phenotype (recessive/dominant) impact on the expression of the adaptive phenotype; too early, the mutant will die and too late the mutant will be outcompeted by mutants that came up earlier.

Along the same lines, the past decade has registered an accumulation of mutations during the life of microbes or somatic cells that cannot be simply attributed to DNA replication-associated mutagenesis. For Instance, *Mycobacterium tuberculosis* is spending most of its time in a non-replicative state, motivating geneticists to estimate its mutation rate as a function of time[49] with a genetic clock producing a similar number of mutations to that observed in fission yeast[27]. In *E. coli*, the stationary phase period is followed by a type of programed cell death and selection of survivors exhibiting the GASP phenotype[50,51] that results mainly from mutations reducing the activity of the sigma S/ sigma38 transcription factor (RpoS), the general starvation/stress-response transcriptional regulator[52,53]. Thus, similarly to the relationship previously proposed for the closely linked mTOR pathway[42,54], the fission yeast S/MAPK pathway plays a function analogous to RpoS in bacteria. This similarity indicates that the stress-response pathways are central to the genetics of quiescence. In bacteria, in addition to regulating the stress-response genes, RpoS also participates in metabolism rewiring to rapidly adapt to new environments[52]. Importantly, the RpoS-dependent response participates in the starvation/stress-induced mutation process at double-strand DNA breaks by inducing error-prone polymerases[55] and down-regulating mismatch repair[56–58], a process that also generates indels (reviewed in ref.[54]). However, many different nutritional stresses induce RpoS but with different mutagenic outcomes[59]. The link between hypomorphic *rpoS* mutations and stress-induced mutagenesis is not clear, but RpoS may itself function as a switch by ultimately being inactivated to limit the effects of mutagenesis associated with the GASP phenotype in an adapted population[60].

In mammals, increasing evidence supports the notion that somatic cells accumulate clocklike mutations as well[61,62]. Recently, using a targeted resequencing approach, hundreds of *NOTCH1* mutations have been found in normal esophageal epithelial cells arising in slow-growing basal cells[63], a third of which are indels. The abundance and clonal feature of these mutations indicate the occurrence of independent genetic events that promote a positive clonal advantage by triggering differentiation and shedding of the neighboring wild-type cells[63,64]. It was proposed that the prevalence of genetic variants accumulating in a tissue-specific manner may contribute to chronic diseases associated with age by generating a tissue microenvironment with either positive or negative impact on tumor progression[65]. Members of the S/MAPK family are also implicated in genetic instability[66], and a recent report shows that p38alpha phosphorylates CtIP[67], a key and early player of double-strand break repair[68]. Finally, future works are needed to explore the emerging idea that starvation/stress-induced mutations may constitute a common

strategy in microbes and eukaryotes to improve adaptive functions[54].

In summary, after a prolonged nitrogen-limited incubation fission yeast cells having acquired a mutation in the S/MAPK pathways will exhibit a survival and growth advantage most likely because a sub-limiting amount of nitrogen is continuously released by the surrounding dying cells, fueling the death of more S/MAPK proficient cells. We consider this behavior of a starving population to be an example of kin selection feeding the steady-state number of cells after periods of growth. Self-destruction programs in microbes are not unprecedented and include GASP, thymine-less death as well as diverse antimicrobial strategies in bacteria[69–71] and have been also reported in yeast and fungi[72]. To conclude, the genetic dynamics of starved cells generate a heterogenous population in which an individual takes over. This situation is observed in microbes as well as in somatic cells during aging and impacts on the resistance to therapies.

## Methods

**Long-term nitrogen starvation experiment**. The unswitchable M-smt0 haploid prototrophic wild-type strain PB1623 (Table 1) was used throughout this study[27]. Cells were grown in minimal medium (EMM[73]) containing 5 g/L of glutamate (34 mM) up to a cell density below $10^7$ cells per mL and shifted to a minimal medium without nitrogen (EMM-N) containing 4% glucose at a starting concentration of $10^6$ cells per mL. The medium was not replaced during the experiment but the water lost by evaporation during the incubation was compensated for by weighing the culture every week. After 1, 2, and 3 months, the glucose concentration was still around 2%. The ability to form colonies was monitored for 3 months by plating aliquots of the cultures on YES plates at various time points.

**Phenotype screening**. Survivors were picked and tested for their sensitivity to a range of temperatures, to chemical agents (2 M sorbitol, 4 mM hydroxyurea, 0.01% SDS, 14 µg/mL Thiabendazole) and to a high concentration of salts (0.8 M KCl and 0.3 M CaCl$_2$) that do not affect the growth of wild-type cells. Drop assays starting with $10^4$ cells per drop with fivefold serial dilutions were spotted onto YES plates containing a selective agent and incubated for 3–4 days at 32 °C. They were subjected to secondary testing using a different batch of the medium, and only those recapitulating the sensitivity were kept. The presence of the mutations in the targeted resequencing experiment was confirmed by plating out 120 cells per pool and scoring for their sensitivity to temperature, calcium, HU, and SDS. The DNA of the sensitive colonies was prepared and used to amplify the relevant target genes. Sanger sequencing of the fragments confirmed the presence of several mutations in the various pools (alleles in red in Fig. 2 and Supplementary Table 5).

**Targeted resequencing**. Three independent 10 mL cultures or one 60 mL culture with the minimum number of cell divisions (<27) required to generate $4 \times 10^6$ quiescent cells per mL (cultures 1–3, and subcultures 1–6) were grown with the smallest possible level of mutation. We isolated and froze 1000 colonies, by pools of 100 colonies after 60 (cultures 0–3 and subcultures 1–6) and 90 days (cultures 0–3) of quiescence (Fig. 2a). Similarly, 1000 colonies were also picked at day 1 from cultures 1–3 and the subculture. To collect a similar number of cells for each colony (~$5 \times 10^6$ cells), colonies of a similar size were taken after 3 days and 4 days for the small colonies. The pools were stored at −80 °C in 20% glycerol. From each pool, approximately $2.5 \times 10^7$ cells were used for standard DNA extraction. Each individual PCR product was purified after separation on agarose gels and quantified. Barcoded adapter primers were added to generate the libraries. To cover the genes of interest, primer pairs were designed (Supplementary Table 7) to generate overlapping PCR fragments of an average size of 420 base pairs. PCR reactions were performed with the Q5 high-fidelity polymerase Master Mix (New England Biolabs) according to the manufacturer's recommendations. Each PCR reaction was performed in duplicate to generate two independent batches making it possible to exclude few variants due to errors occurring during the amplification process. The amount of each PCR product was estimated on agarose gels and adjusted to the lowest concentration. Next, equal volumes of normalized PCR products from each sample were pooled and purified using the QIAGEN PCR purification kit following the manufacturer's recommendations. The analysis workflow is described in Supplementary Methods.

**Fluorescent-activated cell sorting**. $4 \times 10^6$ cells (1 mL) were harvested and resuspended in 1 mL cold 70% ethanol overnight at −20 °C. Cells are washed in 50 mM sodium citrate (pH 7) for 15 min and resuspended in 0.5 mL of 50 mM sodium citrate, containing 0.1 mg/mL RNase A for 2 h at 37 °C. Next, we added 0.5 mL sodium citrate, containing 4 µg/mL of propidium iodide. The analysis was performed on 10,000 cells with a MACSQuant Analyzer (Miltenyi) and MACS-Quantify software, and no gating was applied.

**Co-cultures**. Mutants and wild-type cells were grown separately in EMM prior to transferring to EMM-N. At day 1 of quiescence, the wild-type culture was contaminated with the mutant at 0.1%. The kanamycin-resistance cassette was inserted at the chromosomal locus chr2SPBC31A8.02 to mark the mutants, making them readily identifiable. The proportion of kanamycin-resistant colonies was monitored during nitrogen starvation by plating weekly aliquots of the co-culture on YES plates and YES plates supplemented with kanamycin during the 3 months of the experiment.

**Triggering quiescence exit**. To trigger quiescence exit in the absence of cell division, glutamate was added twice a day in the morning and in the evening, at a starting concentration of 20 µM and reaching a maximum concentration of 200 µM after 10 additions, if none of the glutamate is metabolized. For ammonium chloride (1.6 g/L—amount of nitrogen similar to that of glutamate) and yeast extract (5 g/L), the same dilution and procedure were used. Water was used as control (mock). Cell mass was monitored by OD$_{600}$, cell number by Beckman Coulter Z1 Particle Counter, and CFUs by plating onto YES plates. DNA content was determined on ethanol-fixed cells using FACS. For microscopy, cells were stained with a DAPI-Calcofluor solution (2 µg/mL DAPI, 1 µg/mL p-phenylenediamine, 25 µg/mL Calcofluor white) to visualize the nuclei and the septum formation with a Zeiss Axioplan 2 Imaging M epifluorescence microscope.

**Reporting summary**. Further information on research design is available in the Nature Research Reporting Summary linked to this article.

## Data availability

Sequence data that support the findings of this study have been deposited in BioProject ID with the accession code PRJNA604512 (NIH SRA). The source data underlying Figs. 1a, 4b–f, 5a, b are provided as a Source Data file.

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

## Acknowledgements

R.M. was supported by the Pasteur–Paris Sorbonne Université (PPU) International PhD Program (E515 de Sorbonne Université) and La Ligue Nationale contre le Cancer. This

work was supported by the CNRS, the Institut Pasteur and the ANR-13-BSV8-0018 grant. We thank Julianne Smith for the critical reading of the manuscript.

## Author contributions

Conceptualization, B.A. Software, R.M. and S.G. Formal analysis, R.M., S.F., and S.G. Investigation, R.M., C.D., S.G., and S.F. Data curation, R.M. and S.G. Writing, S.F., S.G., and B.A. Data visualization, R.M., C.D., S.F., S.G., and B.A. Supervision, B.A. Project administration, B.A. Funding acquisition, B.A.

## Competing interests

The authors declare no competing interests.
