## [Peer Review File · Nature Communications]

Reviewers' comments:

Reviewer #1 (Remarks to the Author):

Makarenko and colleagues have cultured quiescent fission yeast cells for several months in medium deprived of nitrogen. They report that during this time mutation arise in the quiescent cells and that these cluster in mediators of the S/MAPK pathway. They then propose that inactivation of the pathway provides and selective advantage to the mutant cells because these can re-enter the cell cycle successfully under very low dose of nitrogen. The authors finally propose a model where mutant cells feed on the traces of nitrogen released by neighbouring cell to promote survival in the culture.

1) This study is intriguing and interesting but I found the manuscript very dense and hard to follow. In particular, the experimental procedure not clearly explained in the main text. For instance, I understand that all the cultures described in 2a are performed in medium without nitrogen? Am I correct? The figure should be clarified and the authors should explain better the experimental set up in the main text before discussing each set of results. This problem is also reflected in some of my comments below.

2) In the author's condition wt cells survive for ~20 days in EMM -N before a sudden crash in viability occurs. This is very different from what has been reported by the Yanagida (30116786) and Bahler (25452419) laboratories where cells survive for months without significant drop in viability. I struggle to understand how this is possible and worry that this could reflect differences in wt strains genetic backgrounds. It would be good to report the genetic difference between the wt strain used in this study and the fission yeast reference sequence. Are there any mutations? Any that could explain the poor viability of their wt in EMM -N? The author probably already have these data and it shouldn't be difficult to add this information to the manuscript.

3) As far as I understand, the model of kin selection proposed in this paper is based slow proliferation of survivors in long term culture. I cannot seem to find direct evidence for the presence of proliferating cells in those cultures though. How does the proportion of septated cells change during a typical 2 months experiments (as on figure 1a for instance)?

4) L52: "Accordingly, when we monitored the sensitivity of the survivors to conditions affecting a broad range of cellular functions (Supplemental Experimental Procedures) we found 30 clones out of 152 (20%) exhibiting a strong phenotype after three months".

Where are the data? Also clarify what a "strong phenotype" is.

5) L55: "Genetic crosses indicated that each phenotype is associated with a single mutated locus".

Could the authors show the data this claim is based on?

6) L62: "Importantly, some of the genes uncovered in the 12 strains were also found mutated in the 36 strains picked randomly (Supplementary Table 2)."

I don't understand. Why is it "important"? What does it tell us? Could the authors expand please?

7) L90: "Between two and three months, the distribution of alleles in each culture is fluctuating. For example, in culture 2 the *sgf73-896+T* allele represents 9.6% of the population at 2 months of quiescence and reaches 43.5% a month later at the expense of *mkh1-640+A*, suggesting clonal interference within the population (Figure 2bc). Conversely, in culture 3 the proportion of *win1* alleles

remains stable over the same time frame.”

I am confused here. How is it possible to have some genotypes increasing while some remain unchanged? Could the author please clarify?

8) The authors find many indels in their set of mutations. What is the mechanism behind this? Aren't indels linked to DNA replication? If yes, how is it compatible with quiescence?

9) Have the authors tried to culture wt or mutant cells in filtered medium coming from quiescent cultures? This would strengthen the claim that the amounts of nitrogen released by dying cells can support proliferation.

Reviewer #2 (Remarks to the Author):

This manuscript entitled “Prolonged quiescence reveals the mitotic potential of mutants in the S/MAPK pathways” try to make the case that spontaneous mutations appear in a WT stationary phase population and that the one affecting the S/MAPK pathways outcompete the WT population by utilizing the nitrogen trace released by the surrounding dead cells.

Although the question of mutations arising in non-dividing cells is of great interest for a large audience, the experiments presented here suffer many caveats. At this stage, data do not support the author's claims. Many critical experiments are required to reach the proposed conclusions. Furthermore, many efforts have to be made to re-write the manuscript in a more straight forward manner that would help the reader to follow the author's thinking. In its present form, data are lost in a complicate word flow blurring the conclusions that can actually be draw from the experiments.

Major points

1-To what extend the experimental design influences the results?

i/ In this study, quiescence establishment is triggered by transferring cells into a nitrogen depleted medium containing a large excess of carbon. Half of the mutations found enriched after long-term stationary phase concern the S/MAPK signalling pathway. Many cross talks between this pathway and the TOR signalling pathway that markedly senses nitrogen availability have been identified. Moreover, the MAPK signalling pathway is central to stress response.

In these experiments, cells are transferred from an EMM to EMM-N medium, which undoubtedly cause both a nitrogen starvation and a cellular stress. Would the same frequency of mutation in the S/MAPK signalling pathway be found following N exhaustion (without stress), or if cells are transferred to water (starvation for all nutrients) or to a medium containing nitrogen but no glucose, or no phosphate? In other words, are the results generalizable to quiescence or are they specific of abrupt nitrogen depletion?

ii/ To easily test the hypothesis of a “re-growth” caused by a potential, yet non measured, residual release of nitrogen by dead cells, the same experiment should be done in a chemostat, in which the medium is controlled and constant.

iii/ What would the authors get if they sequence 36 randomly picked strains when the medium is refreshed and how does it compare to non-refresh conditions.

iv/ “Survivors” are tested for resistance to conditions affecting cell wall integrity (sorbitol, SDS, salts, temperature). This is thus not surprising that most of those strains bear mutation(s) in the S/MAPK signalling pathway. These data befuddled the results. Therefore, I would suggest removing all this part of the manuscript (figure1b, sup table 1) and present only the results obtained for the 36 randomly

picked strains. The authors should also comment the fact that within the 36 randomly picked strains, 11 have no mutation, 3 strains bear intergenic mutations and yet they survive.

2-Are the mutations acquired in “quiescence” or when cells divide?

The key issue in that manuscript is to demonstrate that mutations detected at 2-3 month actually arose in non-dividing cells.

i/ The only clue that mutations arise when the population is no longer increasing (stationary phase) is given fig 2d in which it is reported that the proportion of mutations in the “selected” MAPK signalling pathway genes increase from above 0.1% at day 1 in the initial culture to 19 to 50% in subcultures grown for 2 month. It is stated p6 1125-126 that “because most of the mutations in the subcultures are not the same and because their abundance is different when they are, our data strongly support the model in which the largest part of the mutations arise during quiescence”. Knowing that cells massively died between day 1 and day 60 (fig1a, 2 orders of magnitude), could the mutations be pre-existing in the original population, but undetected by the sequencing method? (especially in light of the fact that cells bearing mutation in the MAPK signalling pathway may have a selective advantage as suggested in fig 3). The authors should present sequencing results at day 1, and make the theoretical demonstration that the cell death observed in stationary phase is not enough to explain the frequency observed at day 60.

ii/ Even if we admit that mutations arise in stationary phase, there is no evidence that mutations actually arise in non-dividing cells (quiescent cells). In fact, the authors claim that after phase II, some cells are dividing. Thus, it could well be that mutations arise in these dividing cells and not in quiescent cells. Changes have to be made in the text and the conclusions soften accordingly.

3-Is there really some cell divisions in a WT population at 60 days?

The proposed model posit that mutants in the MAPK signalling pathway divide in stationary phase utilizing the nitrogen traces released by dead WT cells that lyse. In fig1, the authors should measure for the unrefreshed population from day 1 to day 90: i/ the % of dead cells in the population (and not CFU that measure the ability to divide); ii/ the % of dividing cells (the authors could easily detect new-born cells by doing a pulse of fluorescently labelled concanavalin, or proliferating cells using calcofluor); and iii/the amount of glutamate and other potential nitrogen sources released in the medium by HPIc. These data would lay foundations for the following experiments.

Minor:

4- How do the authors explain that in Fig3a, mutant CFU curves do not display an increased due to mutant cell death and re-division from the released nitrogen trace? This suggest that mutants are not dying in the same proportion that WT cells. Again, measuring dead cells and dividing cells (and not CFU) would help to analyse the results....

5- What are the errors bars in fig1a? Are the measured CFU at 60 days significantly different?

Measuring time points after 90 days would convince the readers. Here only one time point make the difference between the two studied conditions, which is not really convincing.

Reviewer #3 (Remarks to the Author):

In this study, Makarenko et al. examined long-term quiescent cultures of the fission yeast *S. pombe*, and detected frequent mutations in stress-responsive MAPK cascades among survivors. They also found that those MAPK cascade mutants maintained relatively high cell viability in the presence of trace amounts of Glu, a condition proposed to mimic long-term cultures where dying cells provide limited nitrogen source. It was proposed that, in quiescent cultures with a trace of nitrogen source, those MAPK cascades have a function to promote cell death, which ensures survival of a population of

cells with acquired mutations that suppress the MAPK-mediated cell death.

The proposed involvement of the S/MAPK cascades in a cellular strategy to generate a survivor population during quiescence is a novel, interesting concept, but the manuscript and the presented data are not necessarily persuasive. Major issues are summarized below.

Major issues:

1) The manuscript is not very well written and is hard to follow. Here is just one example; it took some time for this reviewer to understand that "57 mutations" (Ins 67-68) is the sum of 21 mutations (In 59) and 36 mutations (In 60). Overall, sentences are not very cohesive, and mixed results and discussions may also contribute to the limited readability.

2) Ins 68-71: "in genes belonging to the S/MAPK pathways and to their downstream targets, Pmc1 the vacuolar calcium transporter, eIF4E/CAP-binding factor Tif452 and Sgf73 coding for a subunit of the SAGA transcriptional complex 18-22."

This reviewer has failed to find in the cited references any clear evidence that Pmc1 and Tif452 are downstream targets of the Pmk1 and/or Sty1 MAPK pathways. It may be more appropriate to eliminate the mutations in these two factors from the S/MAPK module mutation category in the analyses.

3) This reviewer supposes that the title of the paper "Prolonged quiescence reveals the mitotic potential of mutants in the S/MAPK pathways" delineates the phenomena shown in Figure 3b, where the MAPK cascade mutants increased their cell numbers back again after 30 days. However, one concern about this experiment is that additionally acquired mutations may have contributed to the observed growing back – for example, it is possible that the SAPK cascade mutants are more prone to spontaneous, adaptive mutations under the condition.

4) The authors found that the S/MAPK mutants maintained higher viability than wild-type cells when 0.02 mM Glu was added twice a day to the culture (Fig. 3e). This observation itself is interesting, but it is not clear whether this particular experimental condition actually mimics long-term quiescent cultures. What evidence indicates this concentration of Glu is comparable to the nitrogen source derived from dying cells in quiescent cultures?

5) The experiments in Fig. 3e should be controlled by those with no Glu chaser.

Reviewers' comments:

We thank all the reviewers for their comments that helped us design new experiments to improve the quality of the manuscript. We have responded to most of the concerns raised by each of the referees. We are now providing a new version of the manuscript in which the text has an introduction, a results section and a discussion. We hope that these modifications improved the readability of the work.

Reviewer #1 (Remarks to the Author):

Makarenko and colleagues have cultured quiescent fission yeast cells for several months in medium deprived of nitrogen. They report that during this time mutation arise in the quiescent cells and that these cluster in mediators of the S/MAPK pathway. They then propose that inactivation of the pathway provides and selective advantage to the mutant cells because these can re-enter the cell cycle successfully under very low dose of nitrogen. The authors finally propose a model where mutant cells feed on the traces of nitrogen released by neighbouring cell to promote survival in the culture.

1) This study is intriguing and interesting but I found the manuscript very dense and hard to follow. In particular, the experimental procedure not clearly explained in the main text. For instance, I understand that all the cultures described in 2a are performed in medium without nitrogen? Am I correct? The figure should be clarified and the authors should explain better the experimental set up in the main text before discussing each set of results. This problem is also reflected in some of my comments below.

To address this reviewer's concern, we have developed our manuscript by adding the missing information. The experimental conditions in Figure 2a have been described in the text (line 157 and 188) and in the Figure legend, as well as in

detail in Methods (line 443). Figure 2a was also clarified and we added a better description in the legend.

2) In the author's condition wt cells survive for ~20 days in EMM -N before a sudden crash in viability occurs. This is very different from what has been reported by the Yanagida (30116786) and Bahler (25452419) laboratories where cells survive for months without significant drop in viability. I struggle to understand how this is possible and worry that this could reflect differences in wt strains genetic backgrounds. It would be good to report the genetic difference between the wt strain used in this study and the fission yeast reference sequence. Are there any mutations? Any that could explain the poor viability of their wt in EMM -N? The author probably already have these data and it shouldn't be difficult to add this information to the manuscript.

The prototrophic strain used in the Yanagida and Bähler experiments is the 972 h⁻, the reference in our sequencing experiments. Since our progenitor strain was sequenced several hundred times (this work and Gangloff 2017), we provide a new Supplementary Table 1 listing all the polymorphisms uncovered and indicating the accession number of the sequencing. In addition, we followed over a 3-month period, the survival of our progenitor in parallel with the 972 strain in our conditions. The two viability curves are similar, and have been introduced in Figure 1a to indicate that no difference in survival is found and stated in the text (line 121). We also added in Figure 1b the analysis of the DNA content by FACS of PB1623 at 1, 15, 21, 30, 40 and 50 days in quiescence. The results for 972 h⁻ are indistinguishable from those of PB1623. The FACS analysis of 972 is presented below to support our claim.

972 h-

We do not know the reasons for the difference in survival observed with previously published work, but our tests indicate that it depends on experimental conditions rather than on the genetic background. Furthermore, we crossed 972 h⁻ with the *pmk1* mutant in our progenitor genetic background and tested two tetrads for their response to the addition of traces of glutamate. We found that the behavior of the segregants is indistinguishable from that reported in Figure 6 confirming that our observations are independent of the genetic background. We provide the result of the experiment as a supplementary Figure 5 and mention it in the text at line 288.

3) As far as I understand, the model of kin selection proposed in this paper is based slow proliferation of survivors in long term culture. I cannot seem to find direct evidence for the presence of proliferating cells in those cultures though. How does the proportion of septated cells change during a typical 2 months experiments (as on figure 1a for instance)?

The kin selection described concerns the "wild-type" cells that by dying allow the "S/MAPK" mutants to survive. The initial evidence of proliferating cells is supported by the experiments presented in Figure 2b,c. For example, the *sgf73-896* +T allele goes from 9.6% at two months to 43.5% of the population a month

later for a similar number of cells forming colonies (roughly 0.5%) in a culture in which no exogenous nitrogen is provided. This increase can only be achieved through division. We added "indicating cell division" in the text (line 168). Further support for cell division is next provided by the co-culture experiments presented in the new Figure 5b where the best example is *styI* alleles. The CFU ability of these mutants decreases from 5×10^3 CFUs/mL to about 10 CFUs/mL, before increasing to reach roughly 10^6 CFUs/mL (a population larger than the original inoculum) and taking over the culture. This information is shown in Figure 5b and discussed in the text (line 244).

4) L52: "Accordingly, when we monitored the sensitivity of the survivors to conditions affecting a broad range of cellular functions (Supplemental Experimental Procedures) we found 30 clones out of 152 (20%) exhibiting a strong phenotype after three months".

Where are the data? Also clarify what a "strong phenotype" is.

Strong phenotype refers to a sensitivity that is reproducible upon secondary retesting on an independent batch of media. The 30 clones that passed this test are now shown in Supplementary Table 2 in which the sensitivity of the colonies to a relevant drug is shown. We have removed "strong" from the text, as the phenotypes speak for themselves, and have clarified the procedure in the Methods (line 427). The reproducible phenotypes are also observable in Supplementary Figure 1a,b.

5) L55: "Genetic crosses indicated that each phenotype is associated with a single mutated locus".

Could the authors show the data this claim is based on?

We have provided the analysis of 4 tetrads for each of the 30 clones identified by phenotyping (Supplementary Figure 1a,b) as well as for the 6 found among the 36 randomly picked colonies (Supplementary Figure 1c).

6) L62: "Importantly, some of the genes uncovered in the 12 strains were also found mutated in the 36 strains picked randomly (Supplementary Table 2)."

I don't understand. Why is it "important"? What does it tell us? Could the authors expand please?

We just wanted to point out that the same genes were found mutated at a frequency much higher than expected by chance. We agree with the reviewer that this statement is dispensable, and we removed it from the text.

7) L90: "Between two and three months, the distribution of alleles in each culture is fluctuating. For example, in culture 2 the *sgf73-896*+T allele represents 9.6% of the population at 2 months of quiescence and reaches 43.5% a month later at the expense of *mkh1-640*+A, suggesting clonal interference within the population (Figure 2bc). Conversely, in culture 3 the proportion of *win1* alleles remains stable over the same time frame."

I am confused here. How is it possible to have some genotypes increasing while some remain unchanged? Could the author please clarify?

We have rewritten the sentence that now reads (line 166) " For example, in culture 2 the *sgf73-896* +T allele represents 9.6% of the population at 2 months of quiescence and reaches 43.5% a month later, indicating cell division, at the expense of *mkh1-640* +A. Conversely, in culture 3 the proportion of *win1* alleles remains stable over the same time frame. These observations indicate that each mutant behaves individually in the population, and that clonal interference can affect the ability to undergo cell division and impact the fitness of the alleles (Figure 2bc)."

8) The authors find many indels in their set of mutations. What is the mechanism behind this? Aren't indels linked to DNA replication? If yes, how is it compatible with quiescence?

The mechanism behind indels formation is not known. We previously proposed that lesions occurring in quiescence get converted into mutations either in quiescence or when cells resume growth (Gangloff 2017kw). Interestingly, the starvation/stress-induced mutations described in bacteria propose a scenario in which RpoS induces translesion polymerases while downregulating mismatch

repair, a combination that also generates indels (reviewed in Fitzgerald 2017). This point has been addressed in the discussion (line 380).

9) Have the authors tried to culture wt or mutant cells in filtered medium coming from quiescent cultures? This would strengthen the claim that the amounts of nitrogen released by dying cells can support proliferation.

We did not perform this experiment because our study indicates that nitrogen is rapidly metabolized by the cells upon release, as shown in Supplementary Figure 2. At 0.06 mM glutamate, for example, the cells use the nitrogen from the medium to engage DNA replication, indicating that the medium will never contain more nitrogen than that supplied by 0.06 mM glutamate. Instead, we show in the revised manuscript (line 253) that traces of NH_4Cl and yeast extract trigger the same response as glutamate (Figure 6 and Supplementary Figures 3,4), i.e. death of wild-type cells and cell division of the *S/MAPK* mutants.

Reviewer #2 (Remarks to the Author):

This manuscript entitled "Prolonged quiescence reveals the mitotic potential of mutants in the *S/MAPK* pathways" try to make the case that spontaneous mutations appear in a WT stationary phase population and that the one affecting the *S/MAPK* pathways outcompete the WT population by utilizing the nitrogen trace released by the surrounding dead cells.

Although the question of mutations arising in non-dividing cells is of great interest for a large audience, the experiments presented here suffer many caveats. At this stage, data do not support the author's claims. Many critical experiments are required to reach the proposed conclusions. Furthermore, many efforts have to be made to re-write the manuscript in a more straight-forward manner that would help the reader to follow the author's thinking. In its present form, data are lost in a complicate word flow blurring the conclusions that can actually be drawn from the experiments.

We are now proposing an extended version of the manuscript that we hope will allow the reader to follow more easily our train of thoughts.

Major points

1-To what extent the experimental design influences the results?

i/ In this study, quiescence establishment is triggered by transferring cells into a nitrogen depleted medium containing a large excess of carbon. Half of the mutations found enriched after long-term stationary phase concern the S/MAPK signalling pathway. Many cross talks between this pathway and the TOR signalling pathway that markedly senses nitrogen availability have been identified.

Moreover, the MAPK signalling pathway is central to stress response.

In these experiments, cells are transferred from an EMM to EMM-N medium, which undoubtedly cause both a nitrogen starvation and a cellular stress. Would the same frequency of mutation in the S/MAPK signalling pathway be found following N exhaustion (without stress), or if cells are transferred to water (starvation for all nutrients) or to a medium containing nitrogen but no glucose, or no phosphate? In other words, are the results generalizable to quiescence or are they specific of abrupt nitrogen depletion?

A paragraph describing the role of the SAPK pathway in the nitrogen starvation condition in fission yeast is now present in the introduction of the manuscript (line 57). A second paragraph with the relevant references also describes that "stationary phase by glucose or nitrogen exhaustion, as well as starvation by iron, sulfur, phosphate or transfer to water is rarely used to study long-term quiescence in fission yeast because cells die rapidly, suggesting a lack of sufficient storage (line 52). Alternatively, transfer from EMM to EMM-N is routinely used in fission yeast investigations related to long-term quiescence, allowing for an easy and direct comparison with the conditions used here. We also report other studies in the discussion section (line 314) showing that HOG1^{Sty1/p38} mutants were also found enriched in different starvation conditions in budding yeast, supporting similar adaptive traits in different organisms.

ii/ To easily test the hypothesis of a "re-growth" caused by a potential, yet non measured, residual release of nitrogen by dead cells, the same experiment should be done in a chemostat, in which the medium is controlled and constant.

Our study indicates that the nitrogen concentration in the medium is a dynamic process since the nitrogen released by the dead cells is constantly metabolized by viable ones. As shown in Supplementary Figure 2, at 0.06 mM glutamate, for example, the cells use the nitrogen from the medium to engage DNA replication, indicating that the medium will never contain more nitrogen than that supplied by 0.06 mM glutamate. Instead, we show in the revised manuscript that traces of NH_4Cl and yeast extract trigger the same response as glutamate (Figure 6 and Supplementary Figures 3,4), i.e. death of wild-type cells and cell division of the *S/MAPK* mutants.

iii/ What would the authors get if they sequence 36 randomly picked strains when the medium is refreshed and how does it compare to non-refresh conditions.

We recently reported 149 unique mutations in 243 sequenced genomes (picked randomly) after three months of quiescence in refreshed medium (Gangloff 2017). We added this information in the introduction (line 88). We also discuss the genes in the *S/MAPK* pathways that were found in published works on fission and budding yeast (line 311).

iv/ "Survivors" are tested for resistance to conditions affecting cell wall integrity (sorbitol, SDS, salts, temperature). This is thus not surprising that most of those strains bear mutation(s) in the *S/MAPK* signalling pathway. These data befuddled the results. Therefore, I would suggest removing all this part of the manuscript (figure1b, sup table 1) and present only the results obtained for the 36 randomly picked strains. The authors should also comment the fact that within the 36 randomly picked strains, 11 have no mutation, 3 strains bear intergenic mutations and yet they survive.

We followed the reviewer's advice and clearly separated the result of the mutations found in the selected and randomly picked clones in the second paragraph of the results (line 135). The screen for phenotypes on quiescent cells was performed on day 1 and 90, and was intended to show the appearance of phenotypes over time. In the present work, the proportion of strains sensitive to various conditions is considerably higher than anticipated from our previous study

(Gangloff 2017). This result is totally unexpected and motivated the subsequent genetic analysis.

However, we disagree with the suggestion to remove the P strains from the analysis, since no temperature sensitive alleles in genes essential for growth (except a point mutation in *cutI*) were found, it actually reinforces the bias in the selection process observed. In addition, since the proportion of S/MAPK mutants in the R strains (16/36) is not significantly different in the P strains (9/12), ($p=0.095$, $Df=1$ Fisher's exact test, two-sided), we kept the 12 strains in the manuscript and we added in the text: "This elevated level of surviving cells exhibiting a sensitivity phenotype was not expected with the anticipated level of mutation and suggests a strong gain of fitness" line 132. (see Supplementary Table 4).

Concerning the recovery of survivors with no mutation, we have previously established that out of 237 strains sequenced with a sequencing coverage $>50X$, over 130 strains had no detectable mutation (55%) and were following a Poisson distribution (Gangloff 2017). The situation is very comparable in this study (31%). To emphasize this point, we have introduced " When the medium was refreshed every other week to avoid the accumulation of nitrogen released by dead cells, we found a random distribution of mutations that reached 0.6 mutations per genome after three months of quiescence, with about one half of the strains exhibiting no detectable mutation." (line 88).

2-Are the mutations acquired in "quiescence" or when cells divide?

The key issue in that manuscript is to demonstrate that mutations detected at 2-3 month actually arose in non-dividing cells.

i/ The only clue that mutations arise when the population is no longer increasing (stationary phase) is given fig 2d in which it is reported that the proportion of mutations in the "selected" MAPK signalling pathway genes increase from above 0.1% at day 1 in the initial culture to 19 to 50% in subcultures grown for 2 month. It is stated p6 l125-126 that "because most of the mutations in the subcultures are not the same and because their abundance is different when they are, our

data strongly support the model in which the largest part of the mutations arise during quiescence”.

Knowing that cells massively died between day 1 and day 60 (fig1a, 2 orders of magnitude), could the mutations be pre-existing in the original population, but undetected by the sequencing method? (especially in light of the fact that cells bearing mutation in the MAPK signalling pathway may have a selective advantage as suggested in fig 3). The authors should present sequencing results at day 1, and make the theoretical demonstration that the cell death observed in stationary phase is not enough to explain the frequency observed at day 60.

ii/ Even if we admit that mutations arise in stationary phase, there is no evidence that mutations actually arise in non-dividing cells (quiescent cells). In fact, the authors claim that after phase II, some cells are dividing. Thus, it could well be that mutations arise in these dividing cells and not in quiescent cells. Changes have to be made in the text and the conclusions soften accordingly.

Concerning DNA replication generating mutations before nitrogen starvation (i)

Based on our previous publication (Gangloff 2017), the present work provides further evidence that supports the generation of mutagenic lesions in quiescence. We introduced a result paragraph entitled “Mutations occur in quiescence” where the experiment of targeted resequencing of 6 subcultures is described (line 181). As explained in the Methods (line 443), the quiescent culture on day 1 (4×10^7 cells) was obtained by growing a single fresh progenitor for fewer than 27 generations before transfer to G0. Considering the well-established mutation rate of growing *pombe* cells (3×10^{-3} mutation per genome and per generation (Farlow 2015; Behringer 2015), 300 divisions are necessary to generate 1 mutation per genome. Thus, the number of cell divisions generating the quiescent cell population (27 generations) is by far insufficient to generate the diversity and number of mutations found at two months of quiescence. Furthermore, the mutants (Figure 5a,b) display a loss of survival similar or worse than wild-type in the first month of quiescence, indicating that if the mutation is preexisting it would most likely be lost during that period (line 358).

Concerning DNA replication after phase II (ii). The hypothesis that mutations arise during DNA replication following phase II is related to the notion of adaptive mutation that proposes that starved or stressed cells induce mutations.

To clarify these two points, we have discussed them extensively (line 343):

" The molecular mechanism generating mutations during quiescence in fission yeast is not yet understood, but..."

3-Is there really some cell divisions in a WT population at 60 days?

The proposed model posit that mutants in the MAPK signalling pathway divide in stationary phase utilizing the nitrogen traces released by dead WT cells that lyse. In fig1, the authors should measure for the unrefreshed population from day 1 to day 90: i/ the % of dead cells in the population (and not CFU that measure the ability to divide); ii/ the % of dividing cells (the authors could easily detect new-born cells by doing a pulse of fluorescently labelled concanavalin, or proliferating cells using calcofluor); and iii/the amount of glutamate and other potential nitrogen sources released in the medium by HPiC. These data would lay foundations for the following experiments.

We present the colony forming units (CFUs) because our aim is to study the genetics of survivors after long-term quiescence. We have now introduced in Figure 1b the kinetics of nuclear DNA loss measured by FACS. After forty days in quiescence, the population roughly reaches 1% CFUs/ml, with an unmeasurable proportion of cells containing at least 1C DNA, a population size not easily amenable to a reliable quantification by fluorescence microscopy, making the suggested concanavalin A labeling experiment very difficult to interpret. In addition, concanavalin A or halogenic nucleotides are metabolized and provide a nitrogen source to the starved cells.

Two mutually non-exclusive possibilities can explain phase III of the survival curve: i/ the cells have become resistant to death and ii/some cells divide. The new experiments presented in Figure 6 and Supplementary Figure 3 favor the second hypothesis and indicate that, upon addition of traces of glutamate, the mutants engage cell division while the wild-type cells die and accumulate cells that are no

longer stained for nucleic acids and proteins, but bright for Calcofluor white (Supplementary Figure 4). The accumulation of Calcofluor positive cells correlates with the PI negative peak observed in the FACS analysis included in Figure 6 and Supplementary Figure 3 and 4. In addition, the co-culture experiments show that a culture containing around 0.1% (10^4 cells) of *styI* mutant cells sees its *styI* content drop to only 10 cells after 20 days. Four weeks later, they account for over 90% of the cells, a number of cells more than 10-fold above the initial input, fully demonstrating cell division after two months of quiescence (Figure 5b). Finally, concerning the quantification of nitrogen in the medium, our study indicates that it is a dynamic process since the nitrogen released by the dead cells is constantly metabolized by viable ones, as shown in Supplementary Figure 2, and also major point 1/ii of referee #2.

Minor:

4- How do the authors explain that in Fig3a, mutant CFU curves do not display an increased due to mutant cell death and re-division from the released nitrogen trace? This suggest that mutants are not dying in the same proportion that WT cells. Again, measuring dead cells and dividing cells (and not CFU) would help to analyse the results....

Figure 3a is now Figure 5a and shows pure cultures of wild-type and mutant cells. We have added in the text (line 249) " It would also explain why the single cultures of most mutants do not show growth. In this case, nitrogen traces released by dead cells are shared by too many identical mutants, a situation radically different from mutants arising stochastically and progressively."

5- What are the errors bars in fig1a? Are the measured CFU at 60 days significantly different? Measuring time points after 90 days would convince the readers. Here only one time point make the difference between the two studied conditions, which is not really convincing.

In Figure 1a, we showed all the data collected over a period of four years (about 17 long-term quiescence experiments) with different batches of minimum medium and performed by different researchers. We agree that these data confound the message that we want to convey, i.e. there is a change in the slope of phase II in

our experimental conditions. We have removed the published curve in refreshed conditions in which cells keep dying with a constant slope and generate 0.6 mutations per strain with no enrichment in S/MAPK genes (Gangloff 2017). We now show the mean of the 17 individual experiments with an area corresponding to the standard deviation of the means, and included a new set of experiments requested by reviewer 1, performed in parallel with both our progenitor and the 972 h⁻ reference strain. This experiment indicates that the two strains behave alike and confirms the presence of phase III whose slope is reduced compared with phase II. We have updated the figure to make it more readable (Figure 1a) and have added in the legend: " The ability to form colonies (CFUs) was monitored on 17 independent cultures for three months by plating aliquots of the cultures on rich medium at various time points (black curve). The mean is represented and the gray area delineates the limits of the standard deviations. The red (wild-type reference strain 972 h⁻) and blue (our wild-type progenitor PB1623) curves were made in parallel on a culture that was split into three on day 1 of quiescence. The CFUs were determined in triplicate at days 21, 61, 73, 79 and 90."

Reviewer #3 (Remarks to the Author):

In this study, Makarenko et al. examined long-term quiescent cultures of the fission yeast *S. pombe*, and detected frequent mutations in stress-responsive MAPK cascades among survivors. They also found that those MAPK cascade mutants maintained relatively high cell viability in the presence of trace amounts of Glu, a condition proposed to mimic long-term cultures where dying cells provide limited nitrogen source. It was proposed that, in quiescent cultures with a trace of nitrogen source, those MAPK cascades have a function to promote cell death, which ensures survival of a population of cells with acquired mutations that suppress the MAPK-mediated cell death.

The proposed involvement of the S/MAPK cascades in a cellular strategy to generate a survivor population during quiescence is a novel, interesting concept, but the manuscript and the presented data are not necessarily persuasive. Major issues are summarized below.

Major issues:

1) The manuscript is not very well written and is hard to follow. Here is just one example; it took some time for this reviewer to understand that "57 mutations"(Ins 67-68) is the sum of 21 mutations (In 59) and 36 mutations (In 60). Overall, sentences are not very cohesive, and mixed results and discussions may also contribute to the limited readability.

We are now providing an extended version of the manuscript in which the text has an introduction, a result section and a discussion. We hope to have greatly improved the readability of the work.

2) Ins 68-71: "in genes belonging to the S/MAPK pathways and to their downstream targets, Pmc1 the vacuolar calcium transporter, eIF4E/CAP-binding factor Tif452 and Sgf73 coding for a subunit of the SAGA transcriptional complex 18-22."

This reviewer has failed to find in the cited references any clear evidence that Pmc1 and Tif452 are downstream targets of the Pmk1 and/or Sty1 MAPK pathways. It may be more appropriate to eliminate the mutations in these two factors from the S/MAPK module mutation category in the analyses.

We have developed this part by introducing more references, with each gene followed by cognate reference(s) and rephrasing as follows, line 142: " We also identified 36 mutations in the 36 strains sequenced, among which 16 (Supplementary Table 4, Supplementary Figure 1c) are in genes belonging to the S/MAPK pathways (Waskiewicz 1997, Sanso 2011), to their downstream targets eIF4E/CAP-binding factor Tif452 (Ptushkina 2004) and Sgf73 coding for a subunit of the SAGA transcriptional complex (Mason 2017, Kim 2019, Hickman 2011). We also found Pmc1, the vacuolar calcium transporter, that is remotely related to the S/MAPK pathways and whose absence makes cells sensitive to calcium and rapamycin under salt stress (Ishiguro 2013, Cortes 2004, Sugiura 1998, Deng 2006)".

3) This reviewer supposes that the title of the paper "Prolonged quiescence reveals the mitotic potential of mutants in the S/MAPK pathways" delineates the

phenomena shown in Figure 3b, where the MAPK cascade mutants increased their cell numbers back again after 30 days. However, one concern about this experiment is that additionally acquired mutations may have contributed to the observed growing back – for example, it is possible that the SAPK cascade mutants are more prone to spontaneous, adaptive mutations under the condition.

From WGS, we have data for 36 randomly picked colonies, among which 5 affect the SAPK cascade. Only one of these strains contains an additional mutation (Supplementary Table 3). Among the 31 remaining randomly picked colonies, 8 have more than 1 mutation. Although the sample size is small, the two distributions (1/5 and 8/31) are not significantly different (Fisher's exact test, two-sided, Df=1, $p > 0.999$). Along the same lines, in Figure 5b the co-culture experiments show that two different *sty1* alleles and one *win1* mutant exhibit the same kinetics of growth resumption, strongly indicating that they are the drivers of the outgrow phenotype that does not require any additional mutation. Moreover, addition of traces of nitrogen shows that the *sty1*, *sgf73* and *pmk1* mutants start forming septa and divide after 3 days of treatment (day 3, Figure 6) and do not at this point require any extraneous mutation. Furthermore, it is highly unlikely that the strain could acquire an additional driver mutation in just few days.

We have discussed the potential implication of the S/MAPK in the starvation/stress-induced mutagenesis mechanism that potentially increases genetic diversity when cells are poorly adapted to their environment in the context of their analogy with RpoS (line 380).

4) The authors found that the S/MAPK mutants maintained higher viability than wild-type cells when 0.02 mM Glu was added twice a day to the culture (Fig. 3e). This observation itself is interesting, but it is not clear whether this particular experimental condition actually mimics long-term quiescent cultures. What evidence indicates this concentration of Glu is comparable to the nitrogen source derived from dying cells in quiescent cultures?

As stated above for reviewers 1 and 2, we are now providing new experiments showing that anucleated cells are Coomassie-negative indicating that nitrogen is released by the dead cells, as shown in Supplementary Figure 4. We also show that NH_4Cl and yeast extract trigger the same response as glutamate (Figure 6 and Supplementary Figure 3,4).

5) The experiments in Fig. 3e should be controlled by those with no Glu chaser.

We have added the control in Figure 6 and Supplementary Figures 3,4.

REVIEWERS' COMMENTS:

Reviewer #1 (Remarks to the Author):

I thank the authors for their much-improved manuscript. They have now addressed most of my concerns. Regarding my 3rd point, I still believe that it would have been nice to show direct evidence of proliferation (% septated cells as in figure S5b) in the initial experiments and the competitions from figure 5b and not only for the refeeding experiments. This being said I may have been unclear in my initial review or missed something and leave this at the author's discretion.

Reviewer #2 (Remarks to the Author):

The authors have made a great job in clarifying the text and the figures. They have also answered satisfactorily to many points.

Yet, I still have a major semantic issue. Quiescence is a recent attractive field. Therefore it is critical to use the correct vocabulary. I urge the authors to read the seminal review by Gray et al, 2004 in which it is explained that "quiescence" refers to "a cellular state" and as such has to be used only when describing an individual cell. This semantic concept has been commented and acknowledged in many reviews and is now the standard of the field.

In this paper, there is a confusion between "quiescence" and "stationary phase culture/aging culture". The mutations described here are not arising "in quiescence" meaning within the genome of a non-dividing cell (a quiescent cell) but rather in the few cells that proliferate in "aging cultures", which is very different. This is acknowledged by the authors themselves, in both the text and in response to reviewer #1, when they show evidences for the existence of proliferating cells in the long-term culture presented here (fig 2b & c and co-culture experiments).

Thus, the title and the entire text have to be modified accordingly. More specifically all the sentences containing "mutations occur/arise in quiescence" have to be changed by "mutations occur/arise in long term culture" or "in aging culture".

Furthermore, the term "quiescence survivor" has to be removed since by definition, a quiescent cell is a cell that is capable of re-entering the proliferation cycle after a prolonged period of non-proliferation. Thus, all quiescent cells are by definition "survivors".

Reviewer #3 (Remarks to the Author):

The authors may have misunderstood this reviewer's comment 4), "What evidence indicates this concentration of Glu is comparable to the nitrogen source derived from dying cells in quiescent cultures?". In response, the authors pointed out that dead cells are observable in the quiescent cultures and that Glu can be substituted by ammonium chloride or yeast extract in their experiments; however, they have never addressed the concentration issue brought up by this reviewer. The amounts of nitrogen released by dying cells have also been called into question by Reviewer #1's comment 9) and Reviewer #2's Major point-ii, with no straightforward response to them from the authors. This issue should be more critically discussed when the authors present their model.

The revised manuscript has been improved to some extent, but it is still not a pleasant read. Grammatical errors abound, and the meaning and/or logic of some sentences are hard to understand; extensive editing is a must.

REVIEWERS' COMMENTS:

Reviewer #1 (Remarks to the Author):

I thank the authors for their much-improved manuscript. They have now addressed most of my concerns. Regarding my 3rd point, I still believe that it would have been nice to show direct evidence of proliferation (% septated cells as in figure S5b) in the initial experiments and the competitions from figure 5b and not only for the refeeding experiments. This being said I may have been unclear in my initial review or missed something and leave this at the author's discretion.

We did not quantify the septa in long-term starvation (over 1 month), since FACS analysis indicates that only 1% of the cells retain nuclear DNA, consistent with the proportion of colony forming units.

Reviewer #2 (Remarks to the Author):

The authors have made a great job in clarifying the text and the figures. They have also answered satisfactorily to many points.

Yet, I still have a major semantic issue. Quiescence is a recent attractive field. Therefore, it is critical to use the correct vocabulary. I urge the authors to read the seminal review by Gray et al, 2004 in which it is explained that "quiescence" refers to "a cellular state" and as such has to be used only when describing an individual cell. This semantic concept has been commented and acknowledged in many reviews and is now the standard of the field.

In this paper, there is a confusion between "quiescence" and "stationary phase culture/aging culture". The mutations described here are not arising "in quiescence" meaning within the genome of a non-dividing cell (a quiescent cell) but rather in the few cells that proliferate in "aging cultures", which is very different. This is acknowledged by the authors themselves, in both the text and in response to reviewer #1, when they show evidences for the existence of proliferating cells in the long-term culture presented here (fig 2b & c and co-culture experiments).

Thus, the title and the entire text have to be modified accordingly. More specifically all the sentences containing "mutations occur/arise in quiescence" have to be changed by "mutations occur/arise in long term culture" or "in aging culture".

Furthermore, the term "quiescence survivor" has to be removed since by definition, a quiescent cell is a cell that is capable of re-entering the proliferation cycle after a prolonged period of non-proliferation. Thus, all quiescent cells are by definition "survivors".

We tuned down the term quiescence in our manuscript at many places, including the title, as requested. We also removed the term "quiescence survivor". We are fully aware of the outstanding review by Gray et al (2004) and more recent ones stressing the important and under-studied "floating boundaries (entry and exit) of quiescence". However, the "operational" definition of quiescence refers to the situation observed in glucose exhausted rich medium used with budding yeast, and does not apply to nitrogen deprived fission yeast at a controlled concentration of cells, 100-times lower than in the budding yeast experiments. At this concentration, the cells experience the same homogeneous environment. This experimental setting has been

comprehensively defined as quiescence by the pioneering work by Yanagida in fission yeast and extensively used by outstanding colleagues including Bähler and Martienssen.

Concerning the occurrence of mutation in non-dividing cells, based on the work of Yanagida (Mochida 2005, Genes to cells Figure 5) and ourselves (Ben Hassine 2009 EMBO J. Figure 2), it is documented that DNA lesions, either spontaneous or induced can be repaired during quiescence. Like any biological process, DNA repair is not perfect and will make mistakes at a very low frequency and generate mutations in quiescence. On the contrary, if the lesions persist, they will be fixed by DNA replication upon quiescence exit. We have already comprehensively discussed the reasons why mutations are likely arising in non-dividing cells and not in the few cells that proliferate in aging cultures. Indeed, we show that the cells that proliferate are the ones that have acquired a mutation in genes in the S/MAPK pathways. These arguments are clearly stated in the manuscript. Therefore, we disagree with the statement that: the mutations are not arising in a “in quiescence” but rather in the few cells that proliferate in “aging cultures”.

Reviewer #3 (Remarks to the Author):

The authors may have misunderstood this reviewer’s comment 4), “What evidence indicates this concentration of Glu is comparable to the nitrogen source derived from dying cells in quiescent cultures?”. In response, the authors pointed out that dead cells are observable in the quiescent cultures and that Glu can be substituted by ammonium chloride or yeast extract in their experiments; however, they have never addressed the concentration issue brought up by this reviewer. The amounts of nitrogen released by dying cells have also been called into question by Reviewer #1’s comment 9) and Reviewer #2’s Major point-ii, with no straightforward response to them from the authors. This issue should be more critically discussed when the authors present their model.

The revised manuscript has been improved to some extent, but it is still not a pleasant read. Grammatical errors abound, and the meaning and/or logic of some sentences are hard to understand; extensive editing is a must.

We agree that we did not directly address the nitrogen concentration issue, but instead explained at length the reason why in our rebuttal. We have now clarified the reason why we did not measure the nitrogen concentration in our long-term experiments in the result section **Growth Advantage under Nitrogen Starvation**. We have also expanded our rationale for using the given concentrations of nitrogen sources in the result paragraph **Scavenging Behavior and Kin Selection** and softened our discussion of this point (In summary, after a prolonged nitrogen-limited incubation fission yeast cells having acquired a mutation in the S/MAPK pathways will exhibit a survival and growth advantage most likely because a sub-limiting amount of nitrogen is continuously released by the surrounding dying cells, fueling the death of more S/MAPK proficient cells). Finally, the manuscript has been proofread by a native English-speaking colleague.